# Development Approach Model for Automotive Headlights with Mixed Delivery Methodologies over APQP Backbone

**Costel-Ciprian Raicu** [1,*], **George-Călin Seriţan** [2], **Bogdan-Adrian Enache** [2] **and Marilena Stănculescu** [3]

1   Doctoral School of Electrical Engineering, University "Politehnica" of Bucharest, 060042 Bucharest, Romania
2   Department of Measurements, Electrical Equipment and Static Converters, University "Politehnica" of Bucharest, 060042 Bucharest, Romania; george.seritan@upb.ro (G.-C.S.); bogdan.enache2207@upb.ro (B.-A.E.)
3   Department of Electrotechnics, University "Politehnica" of Bucharest, 060042 Bucharest, Romania; marilena.stanculescu@upb.ro
*   Correspondence: costel_ciprian.raicu@stud.electro.upb.ro

**Abstract:** Headlights' development for the automotive industry is gaining a lot of volatility due to frequent changes in features, styling and design, hardware interfaces, and software upgrades required by the OEM, supplier, or new trends in regulations. Standard development models based on V-cycle compliant with CMMI are not responding with reactivity on constant changes. The article proposes an approach based on mixed development strategies over the different core domains with Lean, Scrum, Feature-Driven Development, and VDI to satisfy the APQP milestones, with a proposal of a canvas-type model, the rapid delivery of headlights is portrayed. The efficiency and effectiveness of the model are assessed based on the assumed number of changes for new high-end headlights, based on experience and real cases. A delivery baseline LED-based Headlight development—planned versus actual—chart is presented and explained.

**Keywords:** agile; APQP; headlights; LED; lean

## 1. Introduction

Vehicles and electronic control units continue to increase in complexity [1,2]; the software is allowing the customer to demand new functionalities with higher frequency, and the unpredictability environment forced suppliers towards flexible and value-driven approaches like agile project management [3]. Software products with their given value proposition and malleability open the opportunity to combine Agile with the Lean methodology for the software domain [4]. For software development, Lean is a fuzzy term, especially in the context of Agile methodology, but used to enhance software development processes and scale up Agile [5], more commonly known today as scaled agile framework (SAFe). Manufacturing organizations are driven by world-class performance and the switch towards global economies, the moving target of quality control and the process of improvement with the never-ending scopes, steered them towards lean and agile adoption for enterprise scope [6]. Enterprises are adopting lean to achieve and develop the value stream for eliminating waste, time, and schedule along with agility which is used for market knowledge to exploit profitable opportunities in a volatile marketplace [7].

Advanced Product Quality Planning (APQP) is used for standardized quality management in the automotive industry [8–10], and contains a methodical approach to define and establish the needed measures to ensure the customer expectations for a given developed product; this way, the information can be deployed across the organization between projects with standardized documents to reduce their number and manage the quality [11]. The project planning phase provides the structure for quality and rapid agility of integrating all documents needed according to APQP [12]. The ability for firm guidance and execution over project phases must provide the means for documentation, revision, planning recovery, and vision [13]. Quality standards and project objectives are ensured over the

backbone of APQP, as a good rule of thumb in all automotive suppliers, and linked with the ISO/TS 16949, which helps to focus on the main design scope, planning, and project information review [14].

Product development and product quality plan are supported by APQP to fulfil the customer needs by providing the framework for procedures and techniques, particularly for the automotive industry, focused to define and establish the steps of the development. APQP methodology consists of one pre-planning stage and five concurrent, collaborative sections, with a continuous cyclic process to Plan, Do, Check and Act (PDCA) with the end scope of risk and weakness discovery to reduce or to eliminate potential failures [9]. The procedures are based always on design quality as a first, after manufacturing on quality and satisfying the needs and specifications of the customer.

Headlights for vehicles are one of the most volatile products to develop, they have many constraints as design, optics, mechanics, hardware and software including networking and diagnostics. The body structure of the vehicle is given by the shape of the headlights and are the first products noticed by a possible customer; hence, during the pilot phase, design changes due to styling, optical performance, or mechanics will impact the electronics and will end with an impact for the overall product performance and will delay the start of production (SOP). The current trend in the automotive industry is to move from a linear or wave-type development approach towards customer-centricity and agility deployment to achieve faster viable products to assess during development with waste reduction.

The paper is structured into 4 chapters. Section 1 is focused on the most known delivery methodologies for headlights products with short introductions on their global use. The market trend analysis and benchmarking to introduce the applicability layer according to ISO/TS 16949 standard is presented in Section 2. Furthermore, in Section 3 there is given a new development and delivery model approach for the headlights development life cycle based on the APQP milestones. More, the advantages of keeping the documentation level as required by the ISO9001 and achieving the cross-domain flexibility with possible improvements of values streams are discussed. Section 4 is dedicated to the overall conclusions. The paper ends with a meaningful and up-to-date list of references.

## 2. Current Market Approach

### 2.1. Lean Development

For more revenue, investment increase, and cost reduction, the competitive presence on the rising international market, the lean methodology for manufacturing is working over all parts of the value stream to decrease or reduce the waste. Value streams are activities needed to be planed, ordered, and supplied for a specific product or value within a supply chain [15–19]. Agility in production, just-in-time production, synchronous production, world-class production, and continuous flow are concepts used in contrast with lean production, which helps reduce costs by continuous improvement, so the cost of services and goods are reduced, thereby profits are increased. The purpose is to remove waste or its reduction ("muda", the Japanese word for waste) [20,21] and to achieve a maximum or optimization usage of the tasks. The added value is given from the consumer perspective and drives quality, which equals consumer desire or willingness to pay for the product or the associated service following their viewpoint cycle is reflected in Figure 1.

Lean manufacturing is based on the embodied principles as the tools below:

- Value stream mapping: maps the flow of activities in the manufacturing process to identify obvious areas of waste and obstructions for the flow of value in the process.
- Pull: create products for demand, material is ensured to be pulled instead of pushed through the manufacturing process.
- Just in time: identifies and eliminates non-value-added activities, obstructions in the flow of value.
- Kanban: pull systems in manufacturing by introducing signals between manufacturing cells.

- Load levelling: work to eliminate pile-up of work in progress, create a smooth flow and enable optimal resource utilization.
- Poka-yoke: eliminate manufacturing errors and eliminate the need for rework.
- Single minute exchange of die (SMED): changeover machines rapidly.
- Kaizen: improve the process continuously [22].

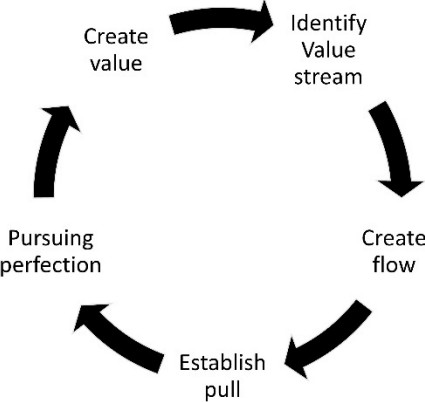

**Figure 1.** Lean Manufacturing principles.

### 2.2. Agile Development

Agile development roots can be traced back to the development of methods like Scrum, Kanban, FDD (feature-driven development) and other iterative solutions for software design and fast delivery like Extreme Programming (XP) or Crystal. These methods comprise various practices, which have their benefits and can be combined. All are based on the Agile Manifesto and included in the so-called "agile methods" to share fundamental commonalities.

The methods are incrementally helping the product development in short iterations, delivering regular product releases to their stakeholders or the end customer, with emphasis on small teams [3,4,13]. The scrum process is portrayed in Figure 2, which shows the iterative character with the circles. The big loop or sprint reflects the short periods in which the product is developed incrementally in small parts and after passing through a reviewed process to check the ready status to be released to the customer or user. The daily meetings, small loop, are kept ensuring the information flow within the team, to update them, and discuss the project progress or current issues [23].

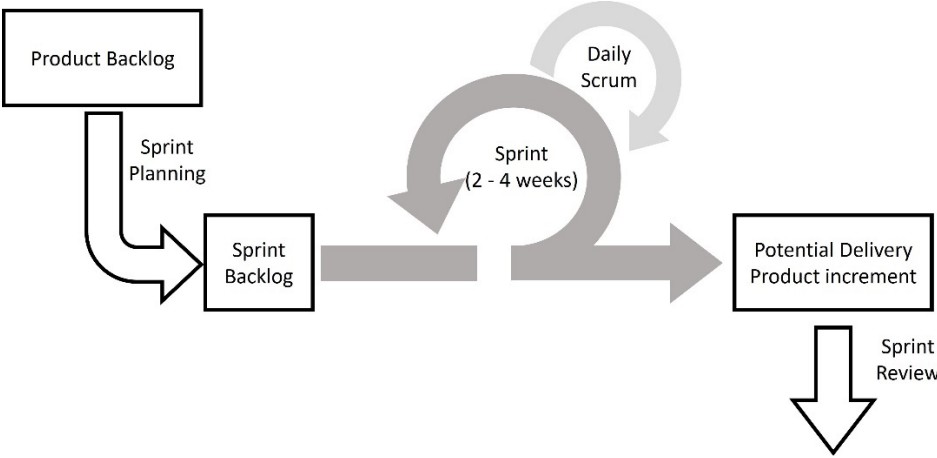

**Figure 2.** Scrum development cycle—part of Agile methodologies.

The agile development adds in the circle of iterations the customer with regular cadence and is proactive in the process to ask the return that is considered during devel-

opment steps. Iteration based development and the acceptance of a lot of changes are considered highly reactive and the perfect match for the product prone to frequent changes. The waterfall model is usually inflexible due to the high level of details in the planning phase; it requires much effort and time to react to any required changes. The standard development process is not involving the customer being inside the circle of the decision nor when the requirements are created for the desired product [24].

Within Agile the Feature Driven Development is a highly adaptive model that is focusing on quality during all development life-cycle and project phases. A feature is a valued function that the end-user wants in the software [25,26], usually can be extended toward system goal as a value delivered, focused mainly on design and delivery phases. Frequent outputs are the expected and testable results that provide progress and status information for the project; the process is shown in Figure 3.

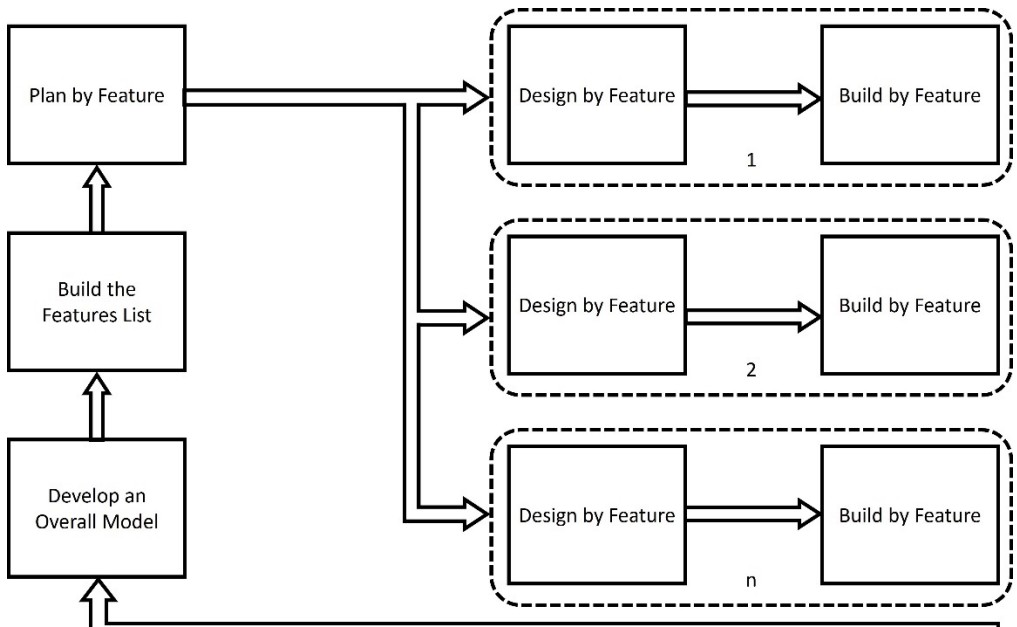

**Figure 3.** FDD development cycle—part of Agile methodologies.

The main targets of FDD for value-added development are its high adaptive development model that is focused on designing and modelling aspects during the early phases of the project for each planned release. Focus on quality is emphasized by the team throughout the development phases [25], feedback can be expected within four weeks by full iteration, which is helping to get fast feedback about the developed product from the customer or end-user.

Integration, software, and feature deliveries are the most important activities that validate the expected results; high flexibility is the key for headlights development and the agile methods are providing the corresponding methodology. Lean and Agile are linked with each other like a balance: on one hand, agile is focused on value-added deliveries, and, on the other hand, lean is focused on waiting until the last critical moment for a decision.

### 2.3. Advanced Product Quality Planning

Advanced Product Quality Planning (APQP) is the development methodology used mainly by the automotive industry to align on a set of procedures and generic templates to reduce the amount of time spent on documentation and to be able to re-use cross-products the documentation when possible [9]. The methodology was created by Chrysler, Ford, and General Motors, USA companies, to provide guidelines for developing a product quality plan that supports the development of a product or service that will satisfy the customer [9–11].

APQP determines the steps necessary to provide a satisfactory product for the customer, based on a structured method centered on product quality planning, divided into five overlapping phases in Figure 4 [8].

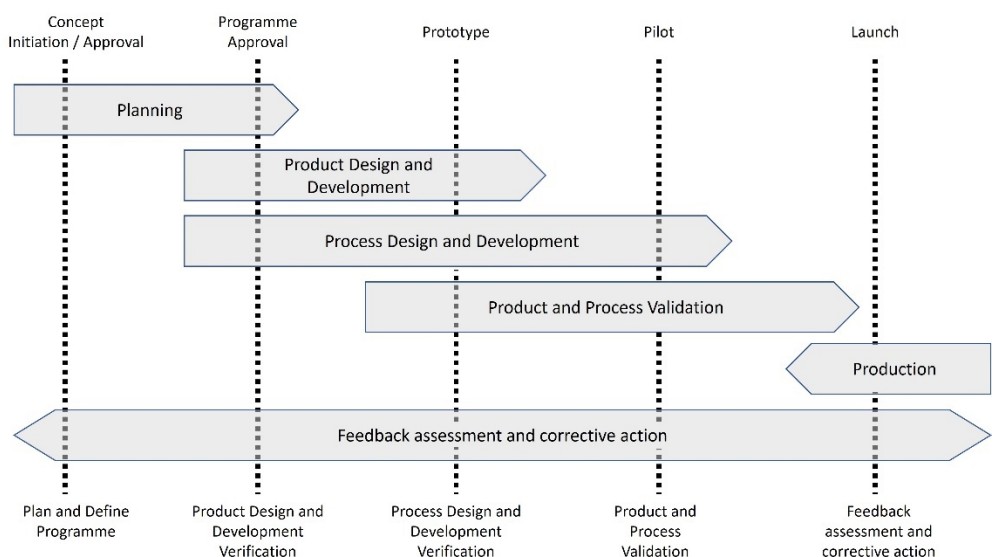

**Figure 4.** APQP phases for quality planning [8].

APQP phases are divided and scoped like [8,9]:

1. Plan and define programme—planning is elaborated based on customer centricity and associated needs to define the common product understanding;
2. Product design and development verification—Design must be frozen and product feasibility assessments with the associated risks are communicated to all stakeholders;
3. Process design and development verification—Manufacturing capabilities are assessed, tooling vendors are nominated based on all product specifications and quality targets, while the cost and volumes desired are kept as a target;
4. Product and process validation—The testing phase for the product and its manufacturing is evaluated to estimate the process capability and the product performance versus the specifications;
5. Feedback assessment and corrective action—Emphasis is on evaluating and improving processes by reducing variations, issues identification, and corrective actions; everything is followed continuously until the product end of life.

The product quality plan is unique for each customer, product, and process. Everything is based on the targeted launch date and implementation of product quality planning [8–11].

Core values generated, from the quality perspective between LEAN and APQP, are linked with the deep desire for quality improvement. We can say that in automotive engineering, both can be satisfied if the foundation of ISO 9001 and ISO/TS 16949 is mastered. The LEAN approach can be considered to survey and provide delivery patterns for on time and on quality delivery, and APQP is focused to ensure transversal documentation and enforcing the production capability and processes (partly also covered in LEAN). APQP helps attain the PPAP of the component with high-quality confidence and Lean will help the MSA improvement and process quality; therefore, Lean-6sigma can be considered much closer to APQP rather than Lean alone.

## 2.4. Headlights Development

Headlights products for automotive industries are mechatronic assemblies, built by suppliers for OEM's, the suppliers are following the standard APQP, whereas the OEM has its proprietary development life-cycle, which in most cases is based on the V-cycle [27].

The V-cycle development life-cycle was initially used in software development and later on adopted for mechatronic components as well, to manage design issues on a micro-level [28], everything encapsulated by the VDI 2206.

The standard product development cycle is following the vehicle development in most cases, the requirements and needs are defined in the project upstream and frozen before a supplier is chosen to manage the convergence and the transition to facilitate their APQP adoption. This process usually is robust and provides strong management and quality control over the product with an engagement to avoid change during development; the baseline is frozen, and the macro-view of the process is shown in Figure 5. The process is a linear one, and its main power resides on agreements and maturity targets over defined quality gates or milestones; after those quality gates are passed, the product may be changed but with a huge impact on quality, timing, and associated costs, due to product maturity and validated tooling, which is to be iteratively rechecked and in some cases changed due to capabilities.

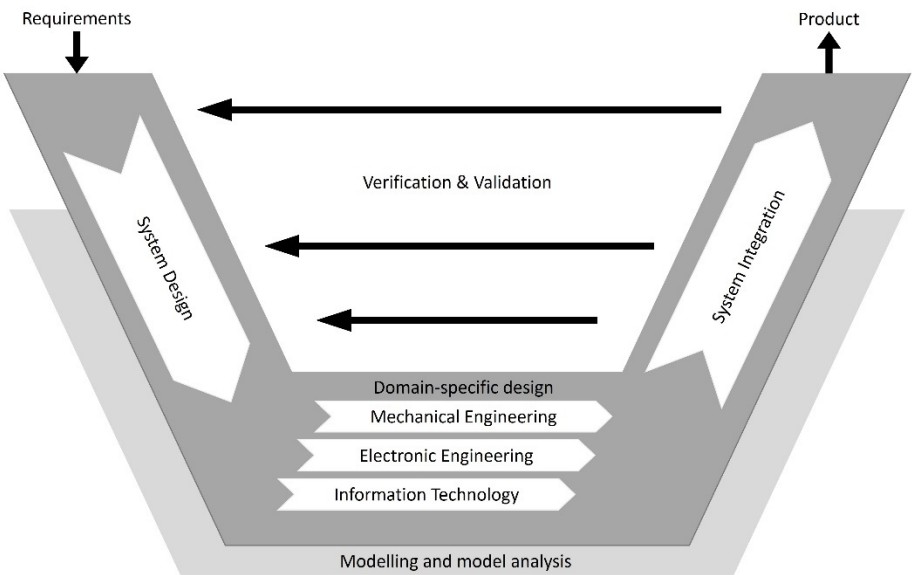

**Figure 5.** V-shaped product development cycle—based on VDI 2206 [28].

The development model based on VDI is a linear development approach with few degrees of flexibility but with high control capability and focused on documentation and deliverables, but less on customer orientation and needs. Customers, the OEM's, must set and avoid any changes on the headlights once the development started; any change will add on delivery delays and cost increases, whereas the other customer-oriented models are involving the customer and are providing the medium to deliver upon its needs.

## 3. Proposed Model

In the Headlight development process, the standard development wave-based application can't sustain a process on the good path, more if it's something new or innovative. Different engineering design activities for headlights and the life-cycles development are shown in Table 1; the frequency use of each approach is noted with high, medium, and low-frequency use, and mapping is based on encountered approaches and quoted literature. Development life-cycles may seem hard to adhere to or converge within a business unit; the domains should focus on one delivery methodology and the project or business unit to set up the applicable bottom layer, for which the end product will be delivered.

**Table 1.** Headlights design approach concerning different development life-cycles.

|  | Lean Development | Agile-Scrum | Agile-FDD | APQP | VDI |
|---|---|---|---|---|---|
| System Design | / | High | High | / | High |
| Electronics | Low | Low | Low | Medium | Medium |
| Mechanics | Medium | / | / | High | Medium |
| Software | Medium | High | High | Low | Medium |
| Styling | Low | / | / | Medium | Low |
| Optics | Low | / | / | / | Low |
| System Integration | Low | High | High | Medium | High |
| Manufacturing | High | Low | / | High | Medium |

Each domain should use different methodologies to address their expectations and value streams, suppliers, or original equipment manufacturer (OEM) will bring their knowledge or experience to prove the efficiency of their model. Usually, they impose the trend, different frequencies being reflected over the number of proposed offers of headlights suppliers across multiple projects during the request for quotation.

Each methodology is used to add value for a domain; they are best on core applications but not confined to them if properly used and scope addressed. Lean on headlights is focused on manufacturing and is met as a standard process within this domain, and is focused not on styling or mechanics delivery nor integration. Its core concepts will add value to reduce waste, decrease timing impacts, and decrease the re-work amount. Agile and Lean are working in a balanced system, software, and features with specific requirements, which are delivered to integration; lean will start when all changes are taken into account and the added value properly addressed. APQP, as a bottom layer, will impose the electronics or hardware maturity with regards to software and mechanics to ensure major deliveries for vehicle integration and customer prototype needs. The methodologies are addressed to ensure a minimum viable product on each milestone of APQP. The headlights development approach is very hard to handle due to exterior impacts by whom they are impacted, body shape re-work, poor optical performance, perceived quality, materials, electronics assumptions, functional errors, and regulation impacts. During the upstream phase, a good path is to set the requirements and global scope of the product, set clear all risks to be taken into account, and plan accordingly. The upstream phase sets up the system strategy and requirements over FDD to increase the scope and full system viability; scrum is used to ensure fast software deliveries and quick bug fixes during integration phases. One big misconception in automotive products or headlights development is the standard application of agile methodologies. This will always fail in the early project stages if wrongly used. They should always start on FDD, set-up system and requirements—managed with scrum and maintained with scrum. APQP will ensure the major delivery expectations.

The development approach or model based on the standard backbone of the APQP, shown in Figure 6, helps to ensure the quality management process used for each engineering activity with the appropriated development processes, which are to have a minimal impact and the minimum cross-impact towards other development activities. Mechanics, styling, and optics have been mainly within the same engineering core, one is hard to be modified without impacting others (except hidden areas); hence, it is necessary to use the same approach for the development cycle. The system design and system integration can be done with different software versions or electronic engineering parts (fast-prototypes), can follow their development cycle, whereas the software another cycle and meet the integration targets converged within a mature product.

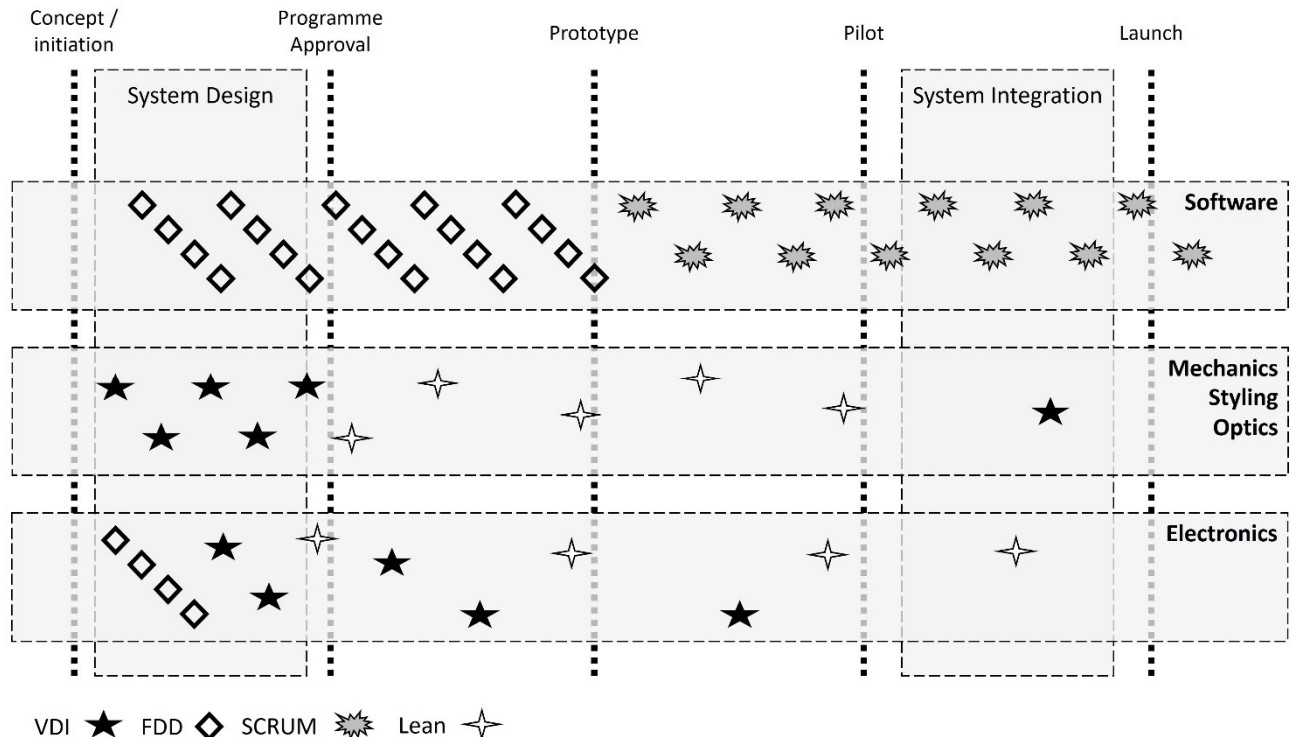

**Figure 6.** Model for LED-based Headlight development.

Between concept/initiation and programme approval, the system design is referred to the requirements, system, electronic, and electrical strategy on a product level with system constraints, first assumptions for style, mechanical design, and optical performance. On each core of competence, the development cycle should be understood as follows:

- Software: Feature Driven Development used until the prototype phase to manage the specification creation, updates, modelling, and simulation as reactive as possible and after to follow up with Scrum updates to cover bug fixes and continuous integration.
- Mechanics, styling, and optics: The design and development approach for this core area must always be focused on iterative targets structured in wave type of development up to programme approval; after, all the changes must be done with a lean strategy and adopted as late as possible—they are the most critical activities with the most impact for the critical path.
- Electronics: They sustain the changes of all the domains and are in some cases quite easy to adapt. During system design, most important are the main features to structure and freeze after the VDI and Lean are adopted: VDI for structure development, tooling, and process design agreement, Lean for flexibility and waste management following the late mechanics, styling, and optics changes.

Development lead-time or the time to market for such products is, usually, between 18 months and up to 24 months. Complexity and degree of re-usability of previous designed concepts or software products are making this feasible. The proposed model helps to reduce the time to market up to 20%, and the main added value is the cost of poor quality reduction by implementing flexible but controllable structures across different engineering domains with a time compression constrained by the company processes and technology [29]. Product development duration is a function of the vehicle development and milestones; sometimes the product may achieve the desired maturity much faster than the vehicle. The pre-concept time duration is not taken into account on an APQP approach. The comparison between the standard APQP duration and the proposed model is shown in Table 2.

**Table 2.** Timing comparison between standard APQP timing duration and the proposed model.

| | Concept to Programme Approval | Programme Approval to Prototype | Prototype to Pilot | Pilot to Launch |
|---|---|---|---|---|
| Standard APQP (months) | 3 | 9 | 6 | 3 |
| Proposed model (months) | 4 | 6 | 4 | 3 |

Fast iterations and customer-oriented development provide the medium for first time right deliveries; adaptability to customer needs and desired features are to be co-designed, which in turn will decrease the internal waste, deliver value faster, and support the integration with the same level of expectations.

A possible drawback of the model may be the organizational change versus the different methods and cross-domain expertise to keep the major milestone on track for ensuring the big picture of the desired product. Validation of the model and the assumption was based on the following formula in Equation (1),

$$\alpha = 1 - \left[ 1 - \frac{100 \times \sum_z z \times \left( \sum_{i,j,k} x_{i,j,k} \times y_{i,j,k} \right) \times t^2}{2} \right] \tag{1}$$

with the:

$t$—reference point for the project duration in months;
$z$—Phases of the development (e.g., pre-concept, prototype, etc.);
$x$—Number of expected releases/loops per phase;
$y$—Expected time spent per each new release, reference assumption, for the project duration in months;
$i$—Software area;
$j$—Mechanics/Styling/Optics area;
$k$—Electronics area.

The equation is predicting the amount of time saved against the expected number of product releases; if alpha is closer to 0 or negative, the process is stable and non-flexible, and up to 25% will be stable and flexible, above 25% product planning is very flexible but un-stable, high risk of failure.

The current model is setting the backbone to provide the dynamics of customer orientation delivery for headlights on a medium for high-quality documentation with iterative processes. Current or previous development methodologies, even if they apply the agile methods, are done only for software areas and are adding on delays and waste for documentation purposes, hardware, and mechanical domains stay on a linear process linked with APQP only in quality and documentation, not relevant for end value delivered. VDI or CMMI, by requiring all data at the beginning of the project launch and will follow up with product change requests for any modifications, are adding waste of value delivery, but they increase the documentation and quality. The proposed model provides the same approach but with the APQP quality and documentation backbone that, when achieved, will ensure the confidence level of the quality management system.

## 4. Discussions and Conclusions

The proposed model facilitates the fast adoption of different life-cycle approaches to attain high flexibility, fast rework, improved reaction time, and clear deliverables for the customer, involved since the early stages of the design. Differences in scopes between different mediums can and will provide some roadblocks; on the other hand, the APQP backbone or other baseline development canvas is the mitigation platform to achieve the results.

The new approach helps delivery, flexibility, and cross-domain convergence for end-user valuation and short lead times, iteratively providing minimum viable deliverables

for feedback, current VDI, ASPICE, or CMMI focused on documentation as a first and a linear approach for all domains with common targets and no deviations. Flexibility, when necessary, for complex headlights with unknown but expected changes along with linear development processes, adds high risks and less opportunity to check the performance or design. The new model is proposing exactly these; one domain can change one point irrespective to another one, which can continue on a parallel track but will cover on APQP target to ensure maturity and quality over expected planning.

Data gathering and requisition were based on LED-based headlights with dedicated controller units to cover the volume of features required by the proposed methodologies. Figure 7 reflects the orientation and data for this model proposed. The headlights data development is covering each APQP milestone with planned and actual delivery required to achieve a minimum viable product. Major iterations were denoted with integer values, whereas the small iterations, which had no cross-domain impact, were denoted with fractions. This was performed for each major activity SW (software), El (electronics), MES (mechanical and optical engineering system) on projects that followed VDI methodologies and Agile methodologies (AgM). X denotes the assumed model expected behaviour. The planned total was added to the graph to reflect the deviation from current projects.

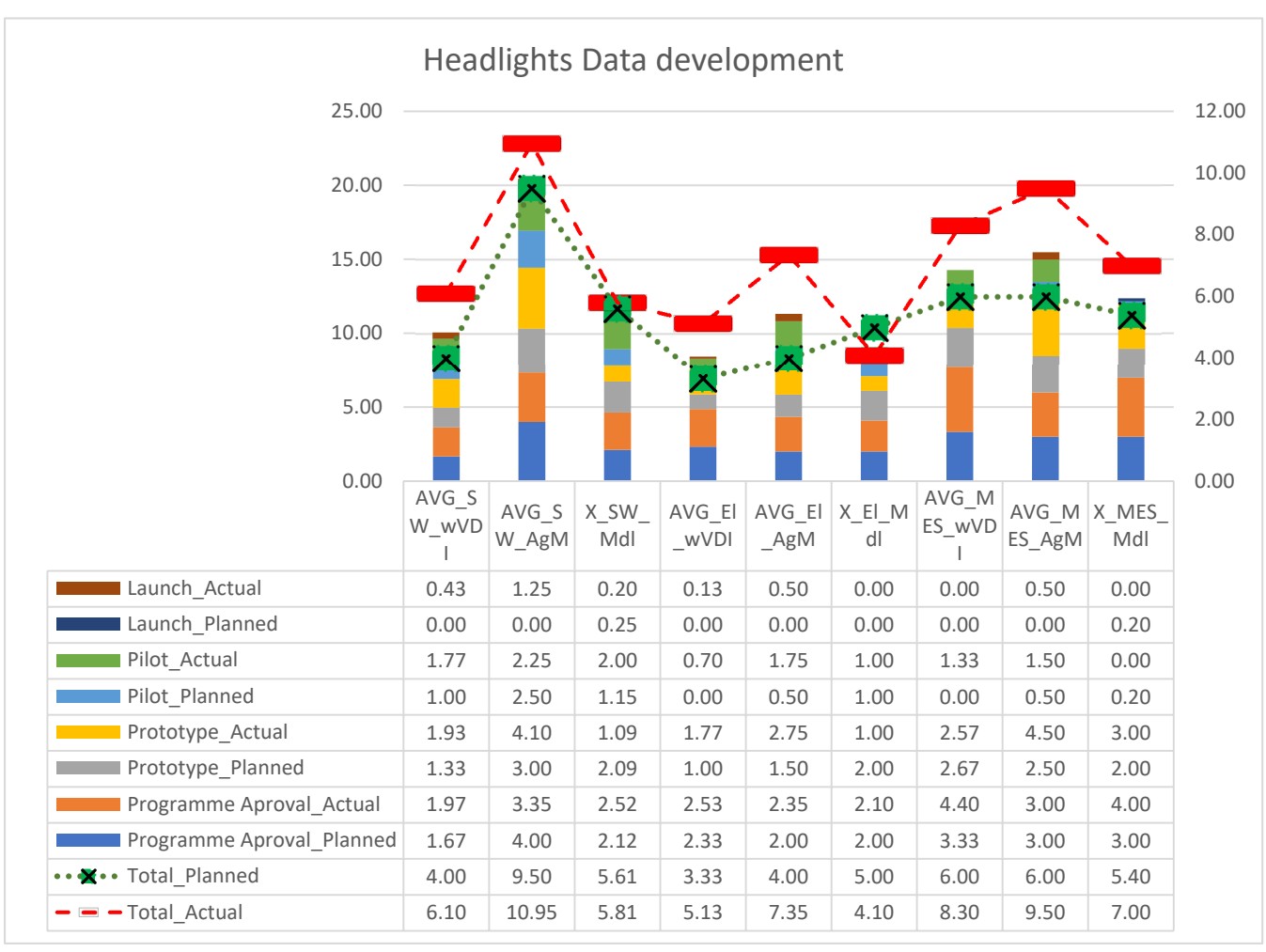

**Figure 7.** Delivery baseline LED-based Headlight development—planned versus actual.

The error concerning the data shown in Figure 7 depends on indirect factors like market trends (which imposed re-designs due to other similar designs) or light signatures (which were launched on market). The new proposed model can function like a white-box type of development where the OEM will take the lead on the application software and

will impose different vendors for different parts of the headlamp. Moreover, the work and the segregation will be achieved through the contract agreement and duly covered by roles and responsibility matrix agreement. Expectations, timings, and iterations irrespective of final methods applied, where they are converged to deliver the work products over a quality development methodology with stable milestones and work with asynchronous development methodology on each domain, will drive, based on company maturity and flexibility, to short delivery timelines.

Development life-cycle based on VDI methodology forced stakeholders on early agreement on project scope and imposed fixed time delivery and fixed expected results, whereas the agile methodology approach encountered on software activities with flexible project scope from suppliers end until the expected maturity will be achieved. VDI and Agile methodology can be similar if the scale is ignored. The first is focused on full project life-cycles (between 18 months to 24 months), and agile on short delivery life-cycles (less than one month); the core differences are the flexibility of change and dynamics of deliveries.

The improved time-to-market of the headlights product, with the proposed model, is achieving lower costs in the engineering effort; with the co-design approach, better quality is delivered. There is no perfect development model, and each headlight, or other products, have their specificity; there may be no feature designers or other constraints such as lack of workforce, all are to be evaluated during the pre-concept phase to be added in the risk registry for choosing the best delivery procedure for implementation.

The efficiency of the model is presumably based on the equilibrium between the total number of changes or releases versus the number of different methodologies used, where we have a small number of changes foreseen with a static model like VDI; any change will have a huge impact, but with a high number of changes scope creep would be un-avoidable.

**Author Contributions:** Conceptualization, C.-C.R.; methodology, C.-C.R.; validation, G.-C.S., B.-A.E. and M.S.; formal analysis, B.-A.E. and G.-C.S.; investigation, C.-C.R. and M.S.; data curation, C.-C.R.; writing—original draft preparation, C.-C.R.; writing—review and editing, C.-C.R. All authors have read and agreed to the published version of the manuscript.

**Funding:** This research was funded by within the program PubArt from Politehnica University of Bucharest.

**Conflicts of Interest:** The authors declare no conflict of interest.

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
