# Peer review of "Development Approach Model for Automotive Headlights with Mixed Delivery Methodologies over APQP Backbone"

_applsci, doi:10.3390/app112210581_

Round 1
Reviewer 1 Report
Although the title is suggesting, it is a bit specific and don't allow to determine the context.
In general, the use of commas, dots, phrases and paragraphs must be reviewed. Try to elaborate direct sentences, avoiding the use of single-phrase paragraphs.
The abstract must be improved. Headlights do no appear. The abstract do not follow the structure recommended by the publisher:
Without headings: (1) Background: Place the question addressed in a broad context and highlight the purpose of the study; (2) Methods: briefly describe the main methods or treatments applied; (3) Results: summarize the article's main findings; (4) Conclusions: indicate the main conclusions or interpretations. The abstract should be an objective representation of the article and it must not contain results that are not presented and substantiated in the main text and should not exaggerate the main conclusions.
References are not ordered. APM is the acronym of IPMA UK, being the largest national association in project management around the world. I suggest to replace the meaning og Agile Project Management.
The use of Agile must be stressed, as well as its combination with Lean.
Relation between APQP, Agile and Lean must be explained.
The meaning of Quality in Lean and APQP approaches differ from the measing in project management contexts, including Agile.
The introduction should briefly place the study in a broad context and highlight why it is important. It should define the purpose of the work and its significance. The current state of the research field should be carefully reviewed and key publications cited. Please highlight controversial and diverging hypotheses when necessary. Finally, briefly mention the main aim of the work and highlight the principal conclusions. As far as possible, please keep the introduction comprehensible to scientists outside your particular field of research.
The structure of the paper should be better justified and explained.
A method section is required.
Originality and novelty must be put in value. The paper must include this, incluying the pertinent discussions with the current literature.
Lean is not described in the context of the application. Idem with agile PM.
Similarities and differences among the three approaches must taken into account. The content of Table 1 must be explained, justified or referenced.
The model is exposed in a prescriptive manner, without leading to the solution, without a previous context. Which is the previous way in which the work is done? The contribution consists of the differences, justifying better results.
Validation must be theoretically justified in order to avoid be a method-driven paper, in which the explanation of the method is the whole.
Discussion of results is missing.
Qualitative and/or quantitative results are expected.
Author Response
Dear reviewer,
First, thank you all for your valuable comments and observation. All of them allowed us to improve a lot our article.
We took into account all your valuable comments. The English language and style have been improved all over the article. The Abstract sections and the introduction were entirely rewritten. Paragraphs were added to follow the received suggestions.
Please find below the answers to your comments.
- Although the title is suggesting, it is a bit specific and don't allow to determine the context.
Thank you very much for your observation. We changed the title: Development approach model for automotive Headlights with mixed delivery methodologies over APQP backbone
- The abstract must be improved. Headlights do no appear. The abstract do not follow the structure recommended by the publisher:
Thank you for your suggestion. We greatly improved the abstract. The modified version is:
Abstract: Headlights development for the automotive industry is gaining a lot of volatility due to frequent changes in features, styling and design, hardware interfaces and software upgrades re-quired by the OEM, supplier or new trends in regulations. Standard development models based on V-cycle and compliant with CMMI are not responding with reactivity on constant changes. The article proposes an approach based on mixed development strategies over the different core do-mains with Lean, Scrum, Feature-Driven Development and VDI to satisfy the APQP milestones, with a proposal of a canvas-type model, the rapid delivery of headlights is portrayed. The efficiency and effectiveness of the model are assessed based on assumed number of changes for new high-end headlights, based on experience and real cases.
- References are not ordered
We re-ordered the entire list of references such that to be cited within the text in an increasing order.
- APM is the acronym of IPMA UK, being the largest national association in project management around the world I suggest to replace the meaning og Agile Project Management
Thank you for your suggestion. Even if there is no specific clear abbreviation agreement for Agile Project Management, we used this abbreviation only once and hence we removed it. Still, in many papers, the APM acronym is quite often used to reflect Agile PM (eg.: PMI org has a published article on their page with this abbreviation used: https://www.pmi.org/-/media/pmi/documents/public/pdf/research/research-summaries/conforto_agile-pm.pdf )
- The use of Agile must be stressed, as well as its combination with Lean
Core concepts are similar but still so different. Agile is focused on the fast delivery of SW products and LEAN is actually oriented on process and efficiency. Agile has short loops whereas LEAN can have extensive loops. We introduced the corresponding comments within the text. For example, we reflected the following inside the larger context: Lean and Agile are linked with each other like a balance, on one hand, agile is focused on value-added deliveries and lean is focused on waiting until the last critical moment for a decision, with scrum or FDD a delivery will be done to integration as soon as is finished but with lean the official introduction will be done only when is necessary and no further changes are expected for a build.
- Relation between APQP, Agile and Lean must be explained
Without previous experience with these processes is almost impossible to stress a good understanding, and this would be our vision which may contradict with someone who is used with APQP only or with the SW delivery in a wave manner. Still bottom-up, imagine the model like a house where the APQP is the foundation LEAN are the walls, VDI the roof and the AGILE (at least in this paper) the mortar which holds everything together
We improved the explanation of the relation between APSP, Agile and Lean. Therefore, we introduced the following paragraph:
Each methodology is used to add value to a domain. Lean on headlights is focused on manufacturing. It is met as a standard process within this domain and it is focused neither on styling or mechanics delivery nor on integration. Its core concepts will add value to reduce waste, decrease timing impacts and decrease the re-work amount. Agile and Lean are working in a balanced system. The software and features with specific requirements are delivered to integration but lean will not start it until all changes are taken into account and the added value properly addressed. APQP, as a bottom layer, will impose the electronics or hardware maturity with regards to software and mechanics to ensure major deliveries for vehicle integration and customer prototype needs. The methodologies addressed to ensure a minimum viable product on each milestone of APQP
- The meaning of Quality in Lean and APQP approaches differ from the measing in project management contexts, including Agile
We can imagine adding the differences in quality between LEAN and APQP, but as core values, from our perspective, they both generate their own paths from a deep desire o quality improvement. More, we can say that in automotive engineering both can be satisfied if the foundation of ISO 9001 and ISO/TS 16949 is already mastered LEAN approach can be considered to survey and provide delivery patterns to deliver on quality and on time. APQP is focused on ensuring transversal documentation and enforcing the production capability and processes (partly also covered in LEAN). APQP helps attain the PPAP of the component with high-quality confidence and Lean will help the MSA improvement and process quality. Therefore, Lean-6sigma can be considered much closer to APQP rather than Lean alone.
- The introduction should briefly place the study in a broad context and highlight why it is important. It should define the purpose of the work and its significance. The current state of the research field should be carefully reviewed, and key publications cited. Please highlight controversial and diverging hypotheses when necessary. Finally, briefly mention the main aim of the work and highlight the principal conclusions. As far as possible, please keep the introduction comprehensible to scientists outside your particular field of research
The introduction was changed and adapted based on the remarks.
- The structure of the paper should be better justified and explained
A. The structure of the paper has been justified as follows:
The paper is structured in 4 chapter. Section 1 is focused on the most known delivery methodologies for headlights products with short introductions on their global use. The market trend analysis and benchmarking to introduce the applicability layer according to ISO/TS 16949 standard is presented in Section 2. Furthermore, in Section 3 there is given a new development and delivery model approach for the headlights development life cycle based on the APQP milestones. More, the advantages of keeping the documentation level as required by the ISO9001 and achieving the cross-domain flexibility with possible improvements of values streams are discussed. Section 4 is dedicated to the overall conclusions. The paper ends with a meaningful and up-to-date list of references
- A method section is required
A. The proposed methodology is part of the presented model of development and if it is extracted from the context. Therefore, we highlighted the proposed method and model.
11. Originality and novelty must be put in value. The paper must include this, incluying the pertinent discussions with the current literature. A. Introduction has been re-written to take into account the received remarks
12. Lean is not described in the context of the application. Idem with agile PM
A. We added some additional data also linked with the above points, still this request is in conflict with a second reviewer comments which actually wants less detailed information, hence we consider the middle ground as the best approach and leave the material like this.
13. Similarities and differences among the three approaches must taken into account. The content of Table 1 must be explained, justified or referenced
A. We explained the data from table 1, in the additional context
- The model is exposed in a prescriptive manner, without leading to the solution, without a previous context. Which is the previous way in which the work is done? The contribution consists of the differences, justifying better results.
A. We explained the VDI process as a whole. VDI, as a full concept, is oriented towards wave-type development for a full product. In the new model, we scaled it down and used it mainly for the upstream phase and electronics. The model can’t be really compared with another one (eg. CMMI, ASPICE, SAFe).
- Validation must be theoretically justified in order to avoid be a method-driven paper, in which the explanation of the method is the whole.
A. We added data and a graph with some of the projects.
- Discussion of results is missing
A. We improved the discussion of the results based on new model behaviour, addressed in section 4.
- Qualitative and/or quantitative results are expected.
We added the following paragraphs for the qualitative/quantitative assessment:
The error with respect to the data shown in the figure 7, depends on indirect factors like market trends (which imposed re-designs due to other similar designs) or light signatures (which were launched on market). The new proposed model can function like a white-box type of development where the OEM will take the lead on the application software and will impose different vendors for different parts of the headlamp. More, the work and the segregation will be achieved by means of contract and dully covered by roles and responsibility matrix agreement.
Development life-cycle based on VDI methodology forced stakeholders on early agreement on project scope and imposed fixed time delivery and fixed expected result. The agile methodology approach encountered on software activities with flexible project scope from suppliers end until the expected maturity will be achieved. VDI and Agile methodology can be similar if the scale is ignored. VDI focuses on full project life-cycles (between 18months to 24months) and Agile focused on the delivery life-cycle (less than one month). The core differences are the flexibility of change and the dynamics of deliveries.
The improved time-to-market of the headlights product, with the proposed model, is achieving lower costs in the engineering effort, with the co-design approach, better quality is delivered. There is no perfect development model and each headlight, or other products, have their specificity, there may be no feature designers or other constraints such as lack of workforce, all are to be evaluated during the pre-concept phase to be added in the risk registry for choosing the best delivery procedure for implementation.
Also, please find attached the work-in document with the corrections aforementioned
Regards,
The authors

Reviewer 2 Report
Agile-Lean Development of LED Headlights over APQP back-bone
“The paper proposes an alternative to the V-cycle development based on the Agile and Lean development ap- 13 proach and to satisfy the APQP milestones. The main criteria tracked is the segregation of development flows based on different domain needs and integration within the structure and fast-prototyping needs for styling validation and acceptance of performance”.
Following, I report the main points:
- It is necessary to explain the novelty of the paper in terms of the methodology
- It is necessary in my opinion to reduce the first 6 pages because the Authors present methodologies known in Literature (it is sufficient a brief description of the different methodologies)
- It not clear the proposed model and the main differences between the old one
I suggest a major revision of the paper before the publication.
Author Response
Dear reviewer,
First, thank you all for your valuable comments and observation. All of them allowed us to improve a lot our article.
We took into account all your valuable comments. The English language and style have been improved all over the article. The Abstract sections and the introduction were entirely rewritten. Paragraphs were added to follow the received suggestions.
Please find below the answers to your comments.
- It is necessary to explain the novelty of the paper in terms of the methodology
We explained the VDI process as a whole. VDI, as a full concept, is oriented towards wave-type development for a full product. In the new model, we scaled it down and used it mainly for the upstream phase and electronics. The model can’t be really compared with another one (eg. CMMI, ASPICE, SAFe).
- It is necessary in my opinion to reduce the first 6 pages because the Authors present methodologies known in Literature (it is sufficient a brief description of the different methodologies
We removed some additional data.
- It not clear the proposed model and the main differences between the old one
We added the following paragraphs for the qualitative/quantitative assessment:
The error with respect to the data shown in figure 7, depends on indirect factors like market trends (which imposed re-designs due to other similar designs) or light signatures (which were launched on market). The new proposed model can function like a white-box type of development where the OEM will take the lead on the application software and will impose different vendors for different parts of the headlamp. More, the work and the segregation will be achieved by means of contract and dully covered by roles and responsibility matrix agreement.
Development life-cycle based on VDI methodology forced stakeholders on early agreement on project scope and imposed fixed time delivery and fixed expected result. The agile methodology approach encountered on software activities with flexible project scope from suppliers end until the expected maturity will be achieved. VDI and Agile methodology can be similar if the scale is ignored. VDI focuses on full project life-cycles (between 18months to 24months) and Agile focused on the delivery life-cycle (less than one month). The core differences are the flexibility of change and the dynamics of deliveries.
The improved time-to-market of the headlights product, with the proposed model, is achieving lower costs in the engineering effort, with the co-design approach, better quality is delivered. There is no perfect development model and each headlight, or other products, have their specificity, there may be no feature designers or other constraints such as lack of workforce, all are to be evaluated during the pre-concept phase to be added in the risk registry for choosing the best delivery procedure for implementation.
We explained the VDI process as a whole. VDI, as a full concept, is oriented towards wave-type development for a full product. In the new model we scaled it down and used it mainly for the upstream phase and electronics. The model can’t be really compared with another one (eg. CMMI, ASPICE, SAFe).
Please find in the attachment, the document reworked
Regards,
The authors

Round 2
Reviewer 1 Report
- Ok.
- Abstract has been improved. However, no specific results are added.
- Ok.
- Ok.
- Ok.
- Ok.
- This explanation should be included in the text.
- The introduction has been only slightly improved. Less than indicated in your comments.
- Ok.
- Ok.
- The introduction has been only slightly improved. Less than indicated in your comments.
- Ok.
- Ok.
- If the model can’t be really compared with another one, then, its novelty must be stressed. However, the model can be partially compared.
- This is no enough.
- Ok.
- Ok.
The paper has been improved.
Author Response
Dear reviewer,
Thank you for all the valuable comments and observations. The article was further improved, following the latest reply's.
Please find below the answers to your comments:
2.2 The abstract has been further improved and specific results were added.
7.2: explanation added in the text Lines 184-192:
Core values generated, from the quality perspective between LEAN and APQP, are linked with the deep desire for quality improvement, we can say that in automotive engineering both can be satisfied if the foundation of ISO 9001 and ISO/TS 16949 is mastered, the LEAN approach can be considered to survey and provide delivery patterns for on time and on quality delivery and APQP is focused to ensure transversal documentation and enforcing the production capability and processes (partly also covered in LEAN). APQP helps attain the PPAP of the component with high-quality confidence and Lean will help the MSA improvement and process quality, therefore, Lean-6sigma can be considered much closer to APQP rather than Lean alone.
8.2 The authors consider that the introduction was adequately improved based on the previous comments of the reviewer. If more details were added we believe that it will become incomprehensible to scientists outside this field of study.
11.2: The authors consider that the introduction was adequately improved based on the previous comments of the reviewer. If more details were added we believe that it will become incomprehensible to scientists outside this field of study.
14.2:
Added the context for the first orientation and set the VDI overview on lines: 213-218
Added a second context guide for the proposed model : lines: 318 – 328
Final comparison added in section 4, on in lines: 336 -344
15.2: Theoretical validation of the data was provided within the context of the article scope. Unfortunately, the level of details is restricted by the NDAs signed by the authors.

Reviewer 2 Report
The paper can be accepted in this new status.
Author Response
Dear reviewer,
Thank you for the positive return and kind positive support, the article was spell-checked and some data was further added to improve its overall content.
Thank you,
The authors
